# Effect of adjuvant radiotherapy in elderly patients with breast cancer

Tanja Nadine Stueber[1]*, Joachim Diessner[1], Catharina Bartmann[1], Elena Leinert[2], Wolfgang Janni[2], Daniel Herr[1], Rolf Kreienberg[2], Achim Woeckel[1], Manfred Wischnewsky[3]

1 Department for Obstetrics and Gynecology, University of Wuerzburg Medical School, Wuerzburg, Germany, 2 Department for Obstetrics and Gynecology, University of Ulm Medical School, Ulm, Germany, 3 Department of Mathematics and Computer Science, University of Bremen, Bremen, Germany

* Stueber_T@ukw.de

**Data Availability Statement:** All relevant data are within the paper.

**Funding:** AW has been funded by the BMBF (Bundesministerium für Bildung und Forschung, Germany) Grant 01ZP0505.https://www.bmbf.de,

## Abstract

### Background

Radiotherapy (RT) is of critical importance in the locoregional management of early breast cancer. Although RT is routinely used following breast conserving surgery (BCS), patients may occasionally be effectively treated with BCS alone. Currently, the selection of patients undergoing BCS who do not need breast irradiation is under investigation. With the advancement of personalized medicine, there is an increasing interest in reduction of aggressive treatments especially in older women. The primary objective of this study was to identify elderly patients who may forego breast irradiation after BCS without measurable consequences on local tumor growth and survival.

### Methods

We analyzed 2384 early breast cancer patients aged 70 and older who were treated in 17 German certified breast cancer centers between 2001 and 2009. We compared RT versus no RT after guideline adherent (GA) BCS. The outcomes studied were breast cancer recurrence (RFS) and breast cancer-specific survival (BCSS). Low-risk patients were defined by luminal A, tumor size T1 or T2 and node-negative whereas higher-risk patients were defined by patients with G3 or T3/T4 or node-positive or other than Luminal A tumors. To test if there is a difference between two or more survival curves, we used the $G^p$ family of tests of Harrington and Fleming.

### Results

The median age was 77 yrs (mean 77.6±5.6 y) and the median observation time 46 mths (mean 48.9±24.8 mths). 950 (39.8%) patients were low-risk and 1434 (60.2%) were higher-risk. 1298 (54.4%) patients received GA BCS of which 85.0% (1103) received GA-RT and only 15% (195) did not. For low-risk patients with GA-BCS there were no significant differences in RFS (log rank p = 0.651) and in BCSS (p = 0.573) stratified by GA-RT. 5 years RFS in both groups were > 97%. For higher-risk patients with GA-BCS we found a significant difference (p<0.001) in RFS and tumor-associated OS stratified by GA-RT. The results remain

The sponsors did not play any role in the study design, data collection and analysis, decision to publish or preparation of the manuscript. The funders did not play any role in this manuscript. TNS received a research grant from Else-Kröner-Forschungskolleg. https://www.ukw.de/behandlungszentren/else-kroener-forschungskolleg/startseite/

**Competing interests:** The authors have declared that no competing interests exist.

the same after adjusting by adjuvant systemic treatment (AST) and comorbidity (ASA and NYHA).

## Conclusions

Patients aged 70 years and older suffering from low-risk early breast cancer with GA-BCS can avoid breast irradiation with <3% chance of relapse. In the case of higher-risk, breast irradiation should be used routinely following GA-BCS. As a side effect of these results, removing the entire breast of elderly low risk patients to spare them from breast irradiation seems to be not necessary.

## Introduction

Breast cancer remains to be the major cancer diagnosis in women. Approximately 65% of the invasive breast cancer patients are aged 40–70 years at initial diagnosis and about 30% of the breast cancer patients are 70 years or older [1]. Survival rates have not improved in elderly patients throughout the years while mortality rates in younger patients have significantly decreased during the last 20 years [2]. The treatment of elderly breast cancer patients differs from the therapeutic approach in younger ones, as elderly patients are prone to geriatric frailty and comorbidities, such as renal failure, liver disease, and/or cerebrovascular disease [3–5]. Moreover, those tumors in elderly patients are often of less aggressive tumor biology.

Postoperative radiotherapy (RT) is the most effective intervention to prevent local relapse after breast conserving surgery (BCS) [6]. In different meta analyses, hazard ratios of 0.2–0.35 have been described meaning that up to eight of ten relapses can be avoided by RT [6–8]. As a result of the improved local tumor control, it has been previously stated that breast cancer related mortality has been significantly decreased during the last decades [6]. Therefore, it has been a general recommendation to perform BCS only in combination with postoperative radiotherapy [9]. If patients cannot or will not accept postoperative RT, mastectomy is the alternative surgical treatment [10]. However, subgroups have been defined that are associated with a low risk for locoregional recurrence (pT1, pN0, R0, HER2-) [11, 12]. In those cases, radiotherapy still adds benefits but can most likely be dispensed even after BCS.

Accelerated partial breast irradiation (APBI) providing radiation therapy to the tumor bed at a higher dose per fraction based on the radiobiologic equivalence is an advisable postoperative approach in properly selected elderly patients, combining advantages of a radical approach that minimizes the risk of undertreatment with efficient reduction of redundant irradiated volume [13, 14]. Similarly, shortened (hypofractionated) dose fraction schedules may be more convenient for older patients [15].

## Methods

### Patients

The BRENDA (breast cancer care under evidence-based guidelines) collective included patients with breast cancer from the Department of Gynecology and Obstetrics at the University of Ulm and from 16 partner clinics in Germany for the period 2001–2009. The exact conditions and inclusion criteria of BRENDA have been described previously [16]. For this retrospective study, we extracted data of 2384 patients aged 70 years and older with primary M0 breast cancer. All patients participated after written and informed consent. If any

information in the database was missing or conflicting, a verification using the original patient file was done. Initial tumor staging and annual follow-up were carried out according to usual recommendations. If the patient was lost to follow-up, data were censored at the date of the last known contact. Adjuvant treatments were checked for guideline adherence. Guideline adherence was defined based on a systematic analysis of the guideline recommendations and statements of the interdisciplinary consensus S3 guideline issued by the German Cancer Society in 2008. A comparison of the treatment recommendations of the S3-guideline and other national breast cancer guidelines from the USA (NCCN, ASCO), Canada (CCO), Australia (NBOCC) and the UK (NICE, SIGN) showed that these guidelines differ only marginally [17]. Recurrence-free survival (RFS) is a composite end point including the following events: Invasive recurrence in the ipsilateral breast or locoregionally, at a distant site, or death from breast cancer.

## Surrogate definition of intrinsic subtypes

To define the intrinsic breast cancer subtypes hormone receptor expression (HR), HER2 expression and cell proliferation marker Ki67 are generally used [18, 19]. As Ki67 was not available in the BRENDA database, we used grading as a surrogate parameter to include the cell proliferation, as described before e.g. by Parise et al. [20], von Minckwitz et al. [21] and Lips et al. [22]. The 5 intrinsic subtypes are defined as follows: Luminal A (HR+/HER2 −/grade1 or 2), luminal B-HER2-negative like (HR+/HER2−/ grade 3), luminal B-HER2-positive like (HR+/HER2+, all grades); HER2-overexpressing (non-luminal, HR−/HER2+) and triple-negative (basal-like, HR−/ HER2−).

## Statistical analysis

Patient characteristics were described with percentages, mean values and standard deviations (SD). When no information was available, the status was coded as missing data. Statistical comparisons for categorical data are carried out using the χ2 test. The distribution of a continuous parameter across a binary variable was tested using the independent-samples Mann-Whitney U test. Survival distributions and median survival times are estimated using the Kaplan–Meier product-limit method. The log-rank test was used to compare survival rates. The Cox proportional hazards model was used to estimate the hazard ratio and confidence intervals. Proportional hazards were tested for all entered variables using statistical and graphical methods (Schoenfeld residuals and log–log plot of cumulative hazard). Confidence intervals for the regression coefficients are based on the Wald statistics. We used hierarchical models to check whether interactions of multivariate significant variables could improve the model.

Furthermore, we additionally tested if there is a difference between two or more survival curves using the $G^p$ family of tests of Harrington and Fleming [23], with weights on each death of $S(t)^p$, where S(t) is the Kaplan-Meier estimate of survival. With rho = 0 this is the log-rank or Mantel-Haenszel test, and with rho = 1 it is equivalent to the Peto & Peto modification of the Gehan-Wilcoxon test.

A non-inferiority log rank test with an overall sample size of 599 low risk patients with GA-BCS (542 in the group with GA-RT and 57 in the group without GA-RT) achieves 80.0% power at a 0.05 significance level to detect an equivalence hazard ratio of 1.40 when the actual hazard ratio is an equivalence hazard ratio of 1.00 and the reference group hazard rate is 0.97. Taking 1.30 instead of 1.40 we need in the low risk group with GA-BCS 911 patients with GA-RT and 96 patients without GA-RT. P-values less than 0.05 were considered statistically significant. Statistical analyses were two sided and carried out using R 3.5, SPSS 26 (IBM) and NCSS 10 for Windows.

## Results

### Basic characteristics

Basic patient characteristics are listed in Table 1. The median observation time was 46 months (mean 48.9±24.8 months). 39,8% (n = 950) were low-risk and 60,2% (n = 1434) were higher-

**Table 1. Basic characteristics of all elderly patients ≥ 70 years included in this analysis.**

| | | Total | low risk | higher risk | asymptotic and exact significance* |
|---|---|---|---|---|---|
| | | 2384 | 950(39.8) | 1434(60.2) | |
| age at diagnosis | | mean: 77.6 (SD 5.6) (median:77.0) | mean: 76.9 (SD 5.1) (median:76) | mean: 78.0 (SD 5.9) (median: 77) | <0.001 |
| | | Range: 70–100 | Range: 70–97 | Range:70–100 | |
| age groups | 70-74y | 847(35.5) | 365(38.4) | 482(33.6) | 0.002 |
| | 75-79y | 717(30.1) | 298(31.4) | 419(29.2) | |
| | ≥ 80y | 820(34.4) | 287(30.2) | 533(37.2) | |
| T-categories | T1 or T2 | 2209(92.7) | 950(100) | 1259(87.8) | < 0.001 |
| | T3 or T4 | 175(7.3) | 0(0.0) | 175(12.2) | |
| grading | G1 | 207(8.7) | 141(14.8) | 66(4.6) | <0.001 |
| | G2 | 1571(66.0) | 809(85.2) | 766(53.4) | |
| | G3 | 602(25.3) | 0(0.0) | 602(42.0) | |
| nodal status | nodal negative | 1367(59.9) | 950(100.0) | 417(31.3) | < 0.001 |
| | 1–3 affected lymph nodes | 512(22.4) | 0(0.0) | 512(38.5) | |
| | >3 affected lymph nodes | 402(17.6) | 0(0.0) | 402(30.2) | |
| intrinsic subtypes | Luminal A | 1561(65.5) | 950(100) | 611(42.6) | < 0.001 |
| | Luminal B/HER2- | 341(14.3) | 0(0.0) | 341(23.8) | |
| | Luminal B/HER2+ | 225(9.4) | 0(0.0) | 225(15.7) | |
| | TNT | 174(7.3) | 0(0.0) | 174(12.1) | |
| | HER2 overexpressing | 83(3.5) | 0(0.0) | 83(5.8) | |
| 100% guideline adherence | non-adherent | 1583(66.4) | 477(50.2) | 1106(77.1) | < 0.001 |
| | adherent | 801(33.6) | 473(49.8) | 328(22.9) | |
| guideline adherent breast conserving therapy (GA-BCS) | carried out—guideline adherent | 1298(54.4) | 654(68.8) | 644(44.9) | < 0.001 |
| | carried out—guideline non-adherent | 85(3.6) | 20(2.1) | 65(4.5) | |
| | not carried out—guideline adherent | 349(14.6) | 62(6.5) | 287(20.0) | |
| | not carried out—guideline non adherent | 652(27.3) | 214(22.5) | 438(30.5) | |
| guideline adherent radiotherapy (GA-RT) | carried out—guideline adherent | 1362(57.1) | 611(64.3) | 751(52.4) | < 0.001 |
| | carried out—guideline non-adherent | 88(3.7) | 21(2.2) | 67(4.7) | |
| | not carried out—guideline adherent | 553(23.2) | 251(26.4) | 302(21.1) | |
| | not carried out—guideline non adherent | 381(16.0) | 67(7.1) | 314(21.9) | |
| adjuvant systemic therapy (AST) | no AST | 440(18.5) | 113(11.9) | 327(22.8) | < 0.001 |
| | hormonal therapy | 1566(65.7) | 818(86.1) | 748(52.2) | |
| | chemotherapy | 141(5.9) | 6(0.6) | 135(9.4) | |
| | hormonal+chemotherapy | 237(9.9) | 13(1.4) | 224(15.6) | |

*The asymptotic significance of the chi-square test and the significance of Fisher's exact test are in our case identical at least to 3 decimal places.

risk. 54,4% patients received GA-BCS of which 85% (n = 1103) received GA-RT and 15% did not obtain GA-RT. 14.6% (n = 349) obtained GA-mastectomy. There was a relatively large number of patients who obtained mastectomy (n = 652; 27.3%), although it was not indicated by the S3-guideline. In the low risk group 29% (n = 276) obtained mastectomy out of which 78% (n = 214) were not GA. Altogether 66% (n = 1583) of the elderly patients were not treated 100% GA (low risk: 50% GA; higher-risk: 77% GA). Low risk patients with GA-BCS but without GA-RT were significantly (p<0.001) older than corresponding patients with GA-RT (median age: 75y vs. 80y). 89.3% of the low-risk patients with GA-BCS were treated with hormonal therapy.

## Recurrence free survival (RFS) and breast cancer-specific survival (BCSS)

There was no significant (log rank p = 0.470) difference in RFS stratified by the 3 age groups (Fig 1). Differences in RFS were significantly different (log rank p<0.001) when stratified by GA-RT in the whole collective of early breast cancer patients ≥ 70 years old (Fig 2). There was a significant difference (log rank p<0.001) in RFS stratified by risk (Fig 3). No significant differences were found when analyzing RFS for low-risk early breast cancer patients stratified by BCS versus mastectomy (Fig 4).

For low risk patients with GA-BCS there was no significant difference in RFS (log rank p = 0.651; Fig 5a) and tumor-associated overall survival (OS) (log rank p = 0.573; Fig 5b) stratified by RT. 5 years RFS for low risk patients with GA-RT as well as for low risk patients without GA-RT were > 97%. In contrast, for higher-risk patients with GA-BCS there were significant differences in RFS (log rank p<0.001; Fig 6a) and in BCSS (log rank p = 0.026; Fig 6b) stratified by GA-RT.

The results do not change with regard to RFS after adjusting by adjuvant systemic therapy (AST) (RFS: low risk p = 0.651; HR = 1.44; 95% CI. (0.33–6.37); higher-risk p = 0.001; HR = 2.86; 95% CI. (1.54–5.31); Fig 7a and 7b).

Additional testing if there is a difference between two or more survival curves using the $G^P$ family of tests of Harrington and Fleming, we also found no significant difference in RFS and in BCSS for low-risk patients with GA-BCS stratified by RT (p = 0.7 and p = 0.6).

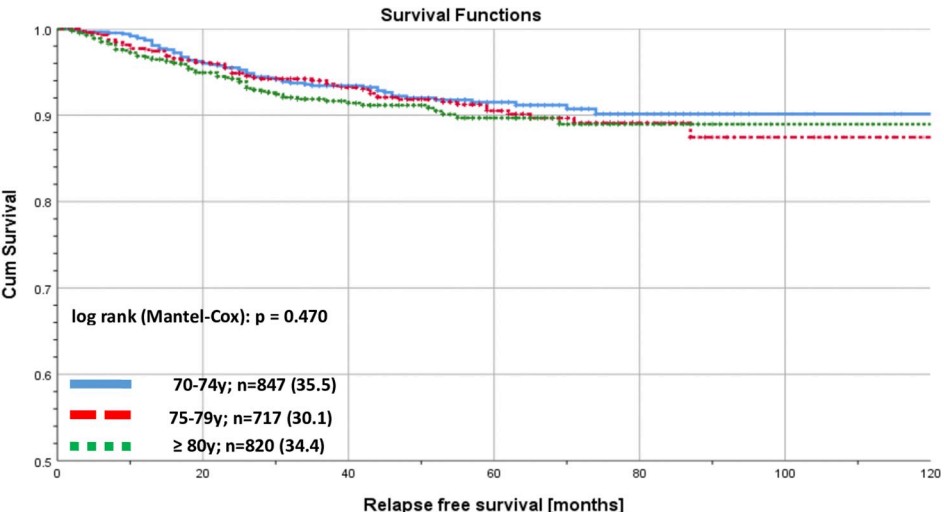

**Fig 1. Recurrence free survival for early breast cancer patients aged ≥ 70 years.** Kaplan-Meier curves of recurrence free survival for early breast cancer patients aged 70 years and older, stratified by age.

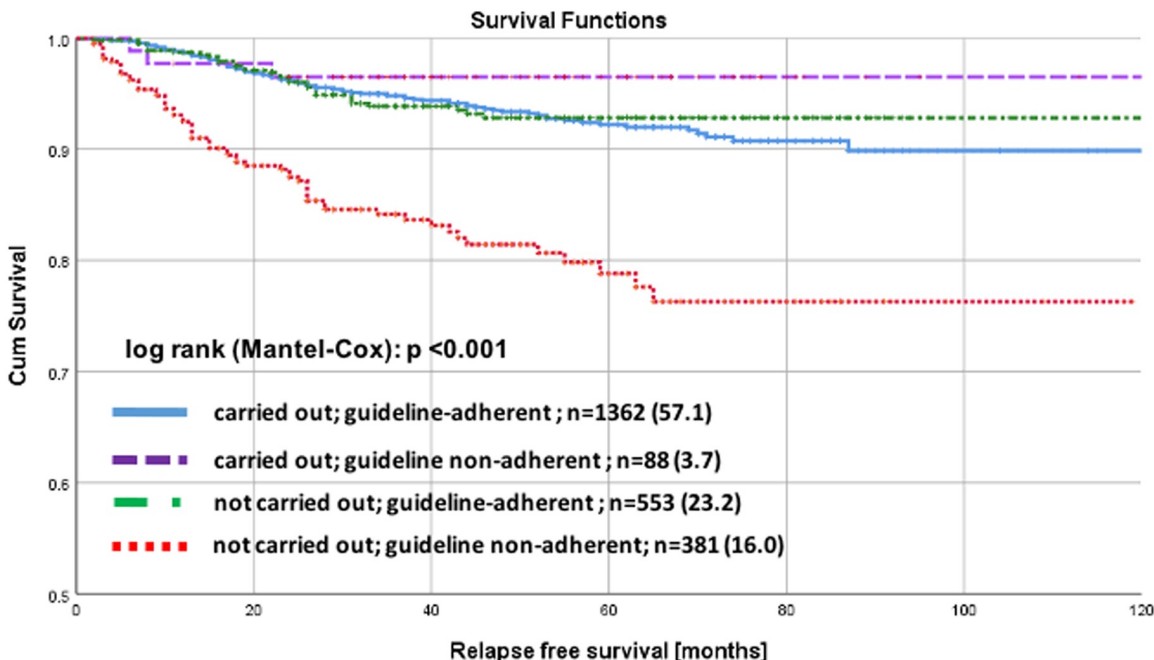

**Fig 2. Recurrence free survival for early breast cancer patients aged ≥ 70 years.** Kaplan-Meier curves of recurrence free survival for early breast cancer patients aged 70 years and older, stratified by guideline-adherent radiotherapy.

Altogether we found, that low risk early breast cancer patients aged 70 and older with GA-BCS without breast irradiation had a chance of relapse of < 3% in 5 years, whereas for elderly higher-risk patients the risk of relapse is up to 9% higher without GA-RT. As a side effect of these results, removing the entire breast of elderly low risk patients seems to have no significant impact on the RFS.

If we restrict our analysis to a similar subgroup of patients as in the CALGB 9343 [8] (age > 70 years with clinical stage I, ER-positive breast cancer treated with lumpectomy

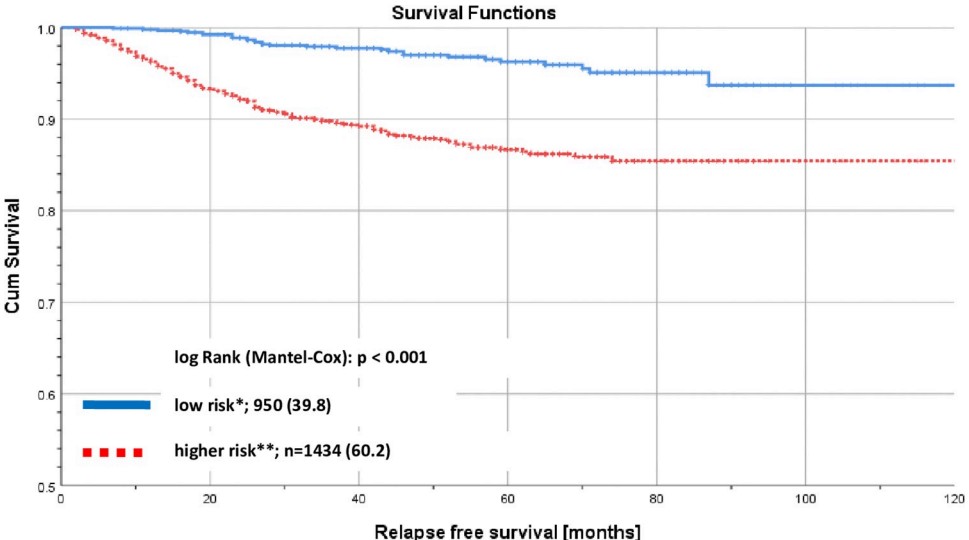

**Fig 3. Recurrence free survival for early breast cancer patients aged ≥ 70 years.** Kaplan-Meier curves of recurrence free survival for early breast cancer patients aged 70 years and older, stratified by risk. Low risk: luminal A and T1/T2 and node-negative. Higher-risk: other than luminal A or G3 or T3/T4 or node-positive.

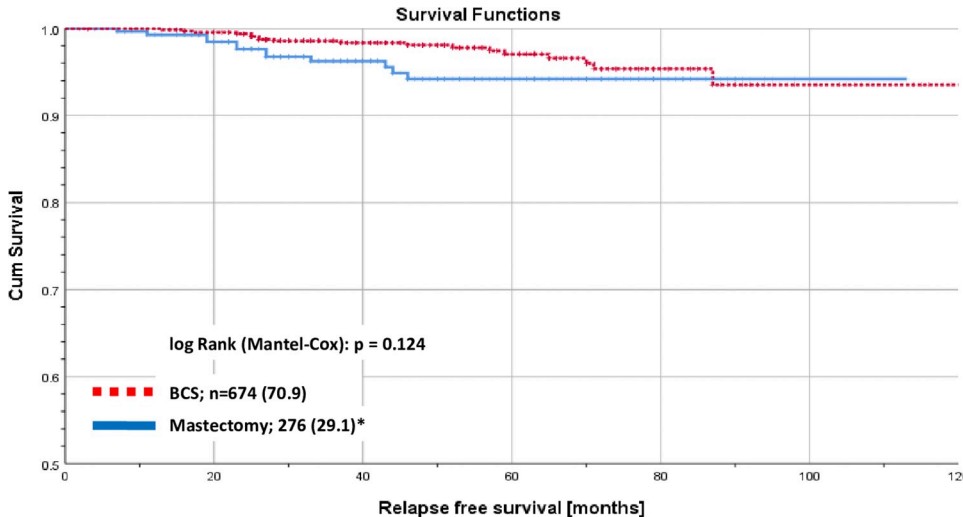

**Fig 4. Recurrence free survival for early breast cancer patients aged $\geq$ 70 years.** Kaplan-Meier curves of recurrence free survival for low risk early breast cancer patients aged 70 years and older stratified by BCS vs. mastectomy. Low risk: luminal A and T1/T2 and node-negative.

followed by hormonal therapy) irradiation adds no significant benefit in terms of recurrence free survival (log rank p = 0.653) and of BCSS (log rank p = 0.662). The CALGB 9343 trial, as well as this study define two types of low risk patients, which both don´t need RT. Therefore, we can combine these two subsets to a new subset of low risk patients defined by (age > 70 years, luminal A and T1/T2 and node-negative treated with BCS) or (age > 70 years with clinical stage I, ER-positive breast cancer, treated with BCS followed by hormonal therapy). For this combined subgroup we have no significant difference in RFS (log rank p = 0.958) and BCSS (log rank p = 0.506).

## Discussion

Our study was based on comprehensive analyses of outcomes of elderly patients with primary breast cancer who did or did not receive breast irradiation following breast-conserving surgery using the multi-center BRENDA Registry. The major goal of the study was to classify those elderly patients who may have no profit of breast irradiation following GA-BCS. The most important findings of the present study were the following:

1. Low-risk early breast cancer patients aged 70 years and older who receive GA-BCS are a subgroup in which breast irradiation seems to add no significant benefit in terms of recurrence free and tumor-associated survival. The chance of relapse within 5 years is <3%.

2. For elderly higher-risk patients, breast irradiation should be used routinely following GA-BCS.

3. Removing the entire breast of elderly low-risk patients to spare them from breast irradiation seems not to be necessary. These patients can obtain BCS without breast irradiation.

In order to reduce morbidity and to increase efficacy of breast cancer treatments, individual tailoring of treatment strategies is needed. Since elderly patients are often presented with several co-morbidities and a wish for de-escalation of therapy, adjuvant therapy components should be reconsidered regarding the expected benefit on the outcome.

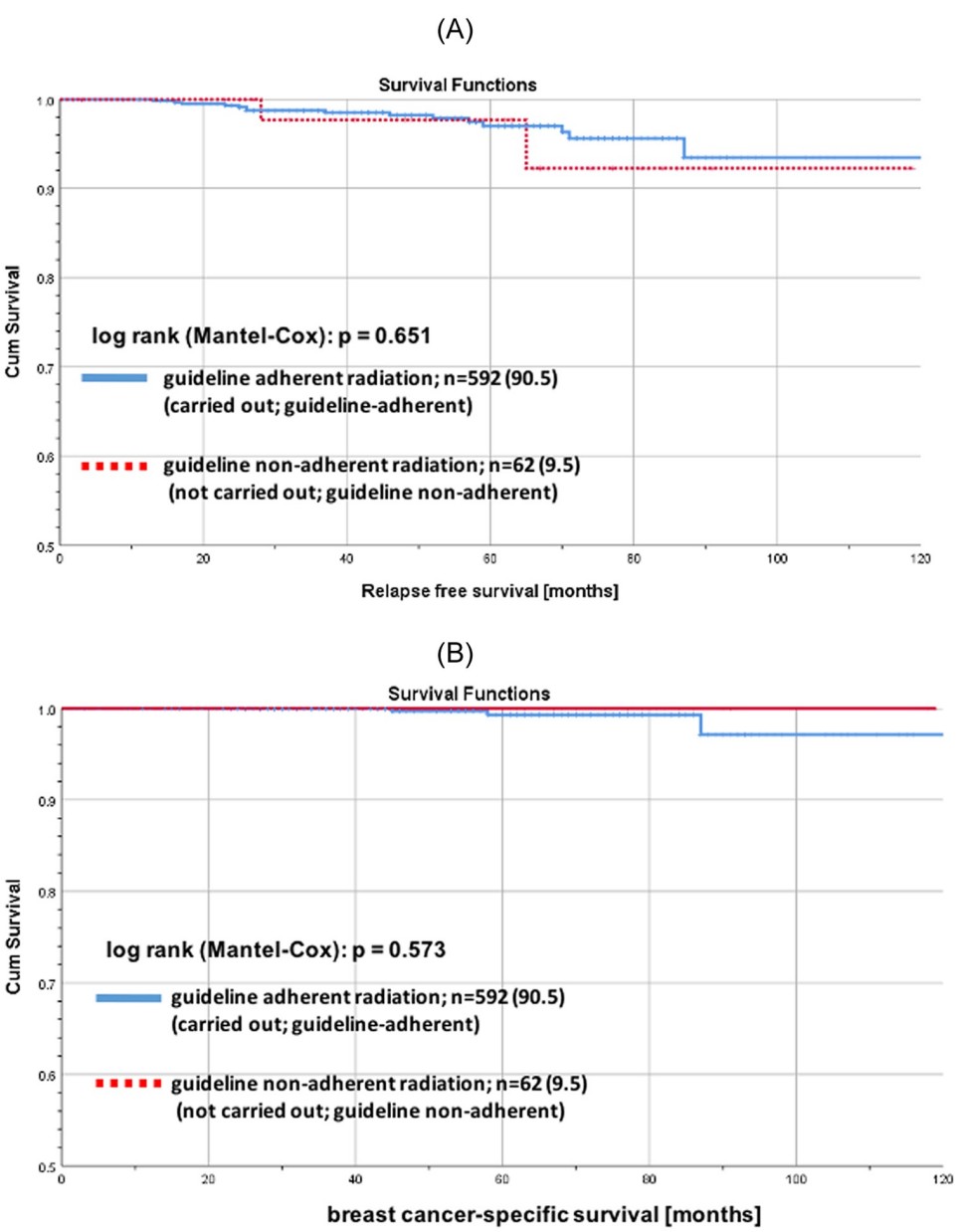

**Fig 5. Recurrence free and tumor-associated overall survival for early breast cancer patients aged ≥ 70 years stratified by guideline adherent radiotherapy.** (A) Kaplan-Meier curves of recurrence free survival for low risk early breast cancer patients aged 70 years and older with guideline adherent BCS stratified by guideline adherent RT. Low risk: luminal A and T1/T2 and node-negative. (B): Kaplan-Meier curves of tumor-associated overall survival for low-risk early breast cancer patients aged 70 years and older with guideline adherent BCS stratified by guideline adherent RT.

In this work, elderly patients received BCS in 58%, while 42% had mastectomy. Interestingly, 652 (27,3%) patients obtained mastectomy even if the guidelines would have recommended to perform BCS while only 85 (3,6%) had BCS instead of mastectomy. This effect has been previously described, showing that elderly patients receive mastectomy more often when compared to younger patients, even if guidelines indicated the contrary [24, 25]. Reasons for

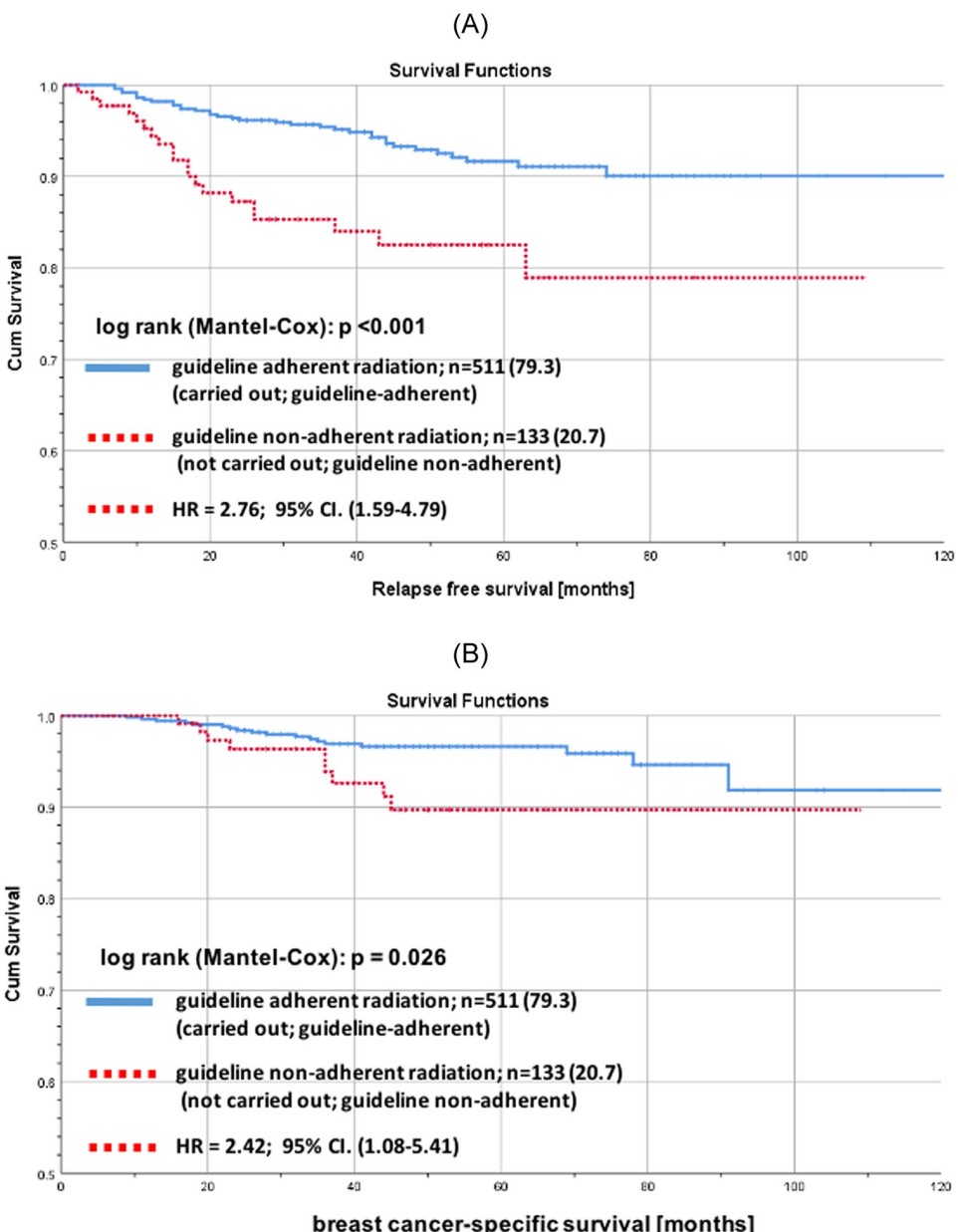

**Fig 6. Recurrence free and tumor-associated overall survival for early breast cancer patients aged ≥ 70 years stratified by guideline adherent radiotherapy.** (A) Kaplan-Meier curves of recurrence free survival for higher-risk early breast cancer patients aged 70 years or older with guideline adherent BCS stratified by guideline adherent RT. Higher-risk: other than luminal A or G3 or T3/T4 or node-positive. (B) Kaplan-Meier curves of tumor-associated overall survival for higher-risk early breast cancer patients aged 70 years or older with guideline adherent BCS stratified by guideline adherent RT. Higher-risk: other than luminal A or G3 or T3/T4 or node-positive.

substandard performance of mastectomies might be the patients wish for final surgical treatment without relevant risk for follow-up resections or local recurrence or physicians'attempt to avoid further adjuvant treatment such as RT. Regardless, mastectomy comes along with relevant morbidity since it has been previously shown that wound infection rates including deep infection and dehiscence occur significantly more often than in patients receiving BCS [26].

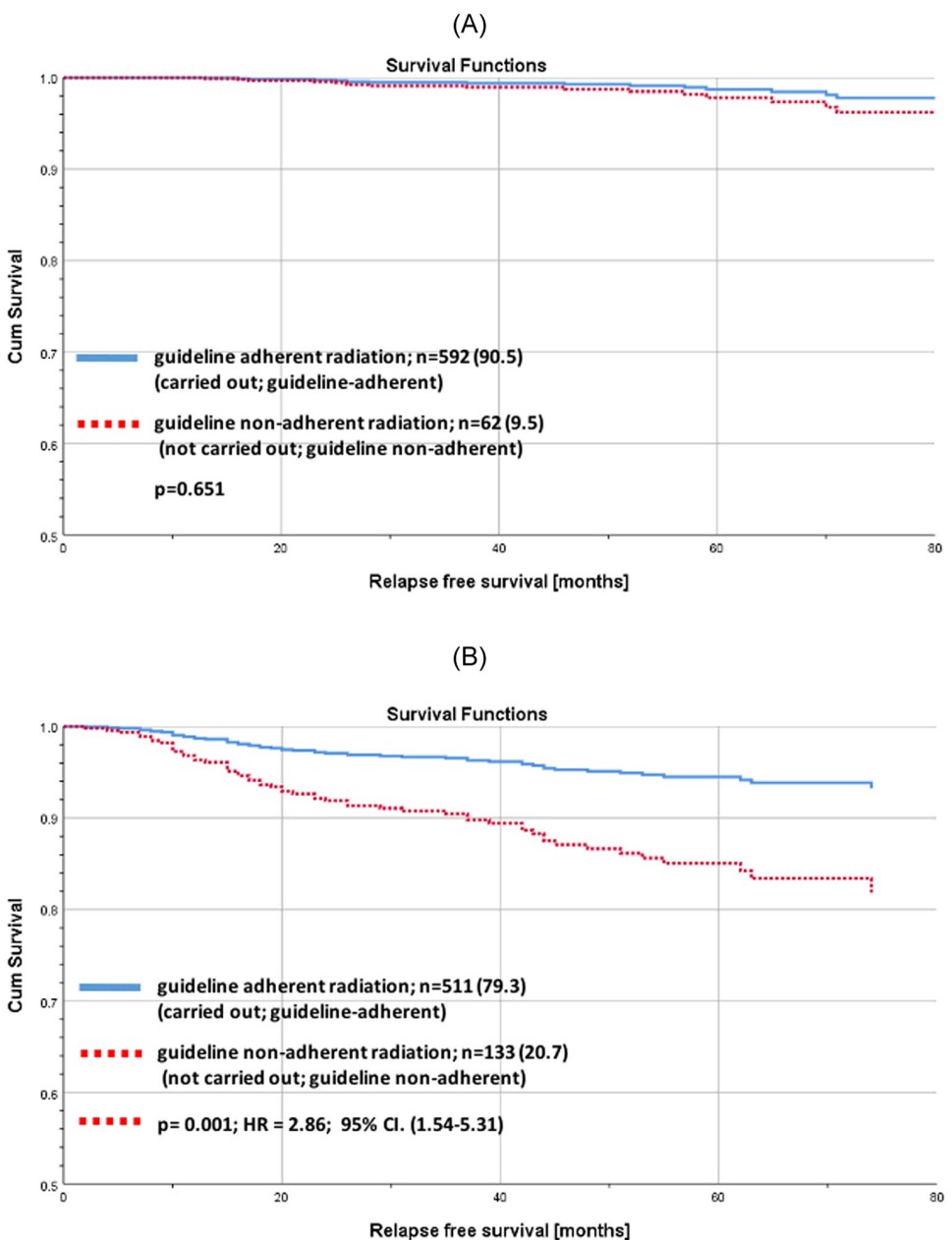

**Fig 7. Cox regression of recurrence free survival for early breast cancer patients aged ≥ 70 years adjusted by adjuvant systemic therapy (AST) and physical status.** (A) Cox regression of recurrence free survival for low risk early breast cancer patients aged 70 years or older with guideline adherent BCS stratified by guideline adherent RT and adjusted by adjuvant systemic therapy (AST), ASA physical status classification system and the New York Heart Association (NYHA) functional classification. Low risk: luminal A and T1/T2 and node-negative. (B) Cox regression of recurrence free survival for higher-risk early breast cancer patients aged 70 years or older with guideline adherent BCS stratified by guideline adherent RT and adjusted by adjuvant systemic therapy (AST), ASA physical status classification system and the New York Heart Association (NYHA) functional classification. Higher-risk: other than luminal A or G3 or T3/T4 or node-positive.

Regarding the survival, we showed that 10-years RFS among elderly patients is in general high and there is no significant difference between patients aged 70–74 years and those older than 80 years. In comparison to older publications survival rates are comparable. Large prospective trials described 10-years RFS rates of around 90% in women diagnosed with early breast cancer between 1994 and 1999 that received endocrine therapy only, while those receiving RT and endocrine therapy showed 98% RFS [8]. Especially, among patients older than 70 years and low-risk early breast cancer, 10-years RFS is excellent, regardless of the different surgical treatments.

For several decades, scientific approaches have been made to analyze if patients could be identified in whom RT might be omitted. All the studies differed conceptually including not only elderly patients but also different tumor sizes or systemic therapies. In summary, no significant differences have been shown regarding OS but some decreases in in-breast recurrences after RT [27–29] as recent meta-analyses confirmed [30, 31]. Interestingly, there is also some evidence that effects of RT also depend on histopathologic subtypes since RT did not improve OS in patients with invasive lobular or uncommon invasive breast cancer types [32].

In our study, we evaluated that in 85% BCS and adjuvant RT were carried out guideline-adherently. The remaining 15% did not receive RT. To evaluate the effect of radiotherapy in the current, prevalent collective of elderly patients we aimed to analyze the different subtypes according to tumor biology. We found that the significantly beneficial effect of RT on RFS was limited to elderly patients with higher-risk early breast cancer. Elderly patients with low-risk early breast cancer instead, showed anyhow excellent RFS and did not significantly profit from adjuvant RT. Also Kunkler et al. stated, that the 5-year ipsilateral breast tumor recurrence rate is probably low enough for women older than 65 years to forgo radiotherapy after BCS [7]. De Boer et al. recently showed that recurrence risk in elderly patients aged $\geq$ 75 years with T1-2N0 breast cancer was low, even without radiotherapy [33].

The results of the long-term follow-up of CALGB 9343 [8] confirm that in the subgroup of women aged > 70 years with clinical stage I, ER-positive breast cancer treated with lumpectomy followed by tamoxifen, irradiation adds no significant benefit in terms of survival, time to distant metastasis, or ultimate breast preservation. In addition to CALGB 9343, we showed the direct difference of effects in a high number of elderly women between higher-risk and low-risk situation. Especially, these results did not change when we stratified by RT and in addition adjusted by adjuvant systemic therapy and comorbidity (ASA and NYHA).

National Comprehensive Cancer Network (NCCN) Guidelines for older women have already recommended omission of RT in elderly patients earlier after the publication of the CALGB 9343 study. Examination of factors associated with a change in RT use in elderly patients with breast cancer showed that age, comorbidity and small tumors were significantly associated with an omission of RT. But the authors also concluded that the use of RT in elderly women was also associated with the treating institution. In this case, the implementation of change in the clinical guidelines showed a wide variety [34]. Moreover, Chu et al. analyzed that RT usage decreased from 71.6% to 67.5% after the publication of CALGB study showing a minimal impact on clinical daily routine concerning RT in older breast cancer patients [35]. In order to improve the implementation of guidelines in these elderly patients protocols including life expectancy estimate and geriatric assessment have been described [36].

## Limitations

Some limitations need to be considered. First, this study has the limitations inherent to a retrospective study. Second, the number of low risk patients with BCS and without RT is small (n = 62). Third, low risk patients with BCS and without RT were significantly older than low risk patients with BCS and RT.

## Conclusions

We conclude that elderly patients could be counselled about indications of limited benefit of adjuvant radiotherapy on the outcome in case of low-risk early breast cancer diagnosis. Elderly patients with higher-risk breast cancer should instead be informed about the beneficial effect of adjuvant RT.

This fact will remain one of several others in order to achieve an individualized treatment strategy for our patients.

## Acknowledgments

We gratefully thank the whole BRENDA study team for their contributions.

## Author Contributions

**Conceptualization:** Tanja Nadine Stueber, Achim Woeckel.

**Data curation:** Achim Woeckel, Manfred Wischnewsky.

**Formal analysis:** Manfred Wischnewsky.

**Investigation:** Manfred Wischnewsky.

**Project administration:** Tanja Nadine Stueber, Achim Woeckel.

**Software:** Manfred Wischnewsky.

**Supervision:** Wolfgang Janni, Achim Woeckel.

**Writing – original draft:** Tanja Nadine Stueber.

**Writing – review & editing:** Joachim Diessner, Catharina Bartmann, Elena Leinert, Wolfgang Janni, Daniel Herr, Rolf Kreienberg, Achim Woeckel, Manfred Wischnewsky.

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
