## [Decision Letter · Decision Letter 0]

5 Nov 2019

PONE-D-19-28656

Effect of adjuvant radiotherapy in elderly patients with breast cancer

PLOS ONE

Dear Dr Stueber,

Thank you for submitting your manuscript to PLOS ONE. After careful consideration, we feel that it has merit but does not fully meet PLOS ONE’s publication criteria as it currently stands. Therefore, we invite you to submit a revised version of the manuscript that addresses the points raised during the review process.

We would appreciate receiving your revised manuscript by Dec 20 2019 11:59PM. To enhance the reproducibility of your results, we recommend that if applicable you deposit your laboratory protocols in protocols.io, where a protocol can be assigned its own identifier (DOI) such that it can be cited independently in the future. For instructions see: http://journals.plos.org/plosone/s/submission-guidelines#loc-laboratory-protocols

We look forward to receiving your revised manuscript.

Kind regards,

Lanjing Zhang, MD, MS

Academic Editor

PLOS ONE

Journal Requirements:

2. We noticed you have some minor occurrence(s) of overlapping text with the following previous publication(s), which needs to be addressed:

https://doi.org/10.1371/journal.pone.0168730

https://doi.org/10.1016/j.clbc.2019.04.015

https://doi.org/10.1093/annonc/mdt539

https://doi.org/10.1186/s12885-016-2345-7

In your revision ensure you cite all your sources (including your own works), and quote or rephrase any duplicated text outside the Methods section. Further consideration is dependent on these concerns being addressed.

3. In ethics statement in the manuscript and in the online submission form, please provide additional information about the patient records/samples used in your retrospective study. Specifically, please ensure that you have discussed whether all data/samples were fully anonymized before you accessed them and/or whether the IRB or ethics committee waived the requirement for informed consent. If patients provided informed written consent to have data/samples from their medical records used in research, please include this information.

Additional Editor Comments:

This is an interesting manuscript, but major revision is needed.

Major points:

-The intrinsic types of breast cancer are not defined according to prior studies (PMID: 29132568).

-Fisher exact test should be used in some subgroups' analyses, instead of Chi-square test.

-Please compare the characteristics of the subgroups that were generated by PSM-IPW grouping. I.e. to test whether PSM-IPW worked.

-How many of your patients probably have used gene based biomarkers (Oncotype or mammoprint)? As recent trials show, they may influenced the patient treatment and outcomes.

Minor points:

-Please cite related articles (PMID: 29132568, PMID: 28746732 and PMID: 31451977)

-Please indicate in the main text, not in the supplement, that the ethical approval has been reviewed and obtained.

Reviewers' comments:

Reviewer's Responses to Questions

**Comments to the Author**

1. Is the manuscript technically sound, and do the data support the conclusions?

Reviewer #1: Partly

Reviewer #2: Yes

2. Has the statistical analysis been performed appropriately and rigorously? 

Reviewer #1: I Don't Know

Reviewer #2: I Don't Know

3. Have the authors made all data underlying the findings in their manuscript fully available?

Reviewer #1: Yes

Reviewer #2: Yes

4. Is the manuscript presented in an intelligible fashion and written in standard English?

Reviewer #1: Yes

Reviewer #2: Yes

5. Review Comments to the Author

Reviewer #1: You state” The treatment of elderly breast cancer patients differs from the therapeutic approach in younger ones, as elderly patients are prone to geriatric frailty and comorbidities, such as renal failure, 94 liver disease, and/or cerebrovascular disease (3-5).”

Please add that the treatment also changes because the cancers are less aggressive (Even within ER +, elderly seem more responsive to endocrine therapy as the percent ER positivity is higher.)

You over state the benefits of RT:

“After evaluation of different subtypes of breast cancer, it is obvious that even those with less aggressive molecular types profit from radiotherapy. In different meta analyses, hazard ratios of 0.2-0.35 have been described meaning that up to eight of ten relapses can be avoided by RT (6-8). As a result of the improved local tumor control, it has been previously stated that breast cancer related mortality has been significantly decreased during the last decades (6). Therefore, it has been a general recommendation to perform BCT only in combination with postoperative radiotherapy (9). If patients cannot or will not accept postoperative RT, mastectomy is the alternative surgical treatment (10). However, subgroups have been defined that are associated with a low risk for recurrence (pT1, pN0, R0, HER2-) (11, 12). In those cases, radiotherapy still adds benefits but can most likely be dispensed even after BCT.”

No randomized trial has shown a survival benefit to RT in early breast cancer.

Always state “locoregional” recurrence, as RT has no impact on distant recurrence in randomized trials.

You never defined intermediate vs high risk or separated out those groups. These need to be defined.

You combine intermediate and high risk into a single group without showing that they act similarly. You need to describe why you made this combination.

In some places you state high risk benefits and in others that intermediate or high risk benefits, but I do not see these differentiated in your data

Your intermediate group likely contains patients that fit the randomized trials that showed RT has not beneficial. (Such as a 1 cm Grade 3 cancer.). If you are using retrospective data to invalidate the conclusions of a randomized trial, please state why you think this is defensible.

When you state that limiting your analysis to 9343 criteria, are you including intermediate and high risk patients who meet that criteria? If so, does that negate your statement that intermediate and high risk need RT?

Reviewer #2: The manuscript here is technically sound and adds additional important information to the literature using a relatively large database of elderly women with breast cancer. The manuscript could be simpler and clearer and my suggestions are below:

1. The use of the terms BCT with and without RT is confusing. If I’m not mistaken breast conserving therapy without radiotherapy should be referred to as BCS (breast conserving surgery). BCT implies the use of radiotherapy.

2. I generally disagree with intro lines 102-104. There has been enough data on low risk patients with small tumors having low recurrence rates without radiotherapy that these statements do not apply in 2019.

3. I’m not sure why the authors separated the groups out by low and intermediate/high. They define the patients that they place in this intermediate/high group but never define the difference between intermediate and high. This should be defined at some point.

4. There are a great deal of words in the paragraph called “basic characteristics” and it is a difficult read. It is essentially explaining Table 1. The paragraph could be started by simply stating that “Basic patient characteristics are listed in Table 1” and then just point out a couple of important distinctions. I do not feel this large paragraph is necessary.

5. I find the terms throughout the manuscript of guideline adherent and guideline non-adherent confusing. One could simply state that guidelines state that radiotherapy is indicated and compare the difference between those having it and those not having it. It also makes the Figures showing relapse free and overall survival confusing.

6. Discussion could be much simpler. Prime 2 and CALGB trials are referenced early on and then a paragraph is allotted to each one later in the discussion. I would just acknowledge each randomized control trial once and shorten discussion.

7. Overall useful information particularly regarding the higher risk group but could be more simply and clearly presented.

6. PLOS authors have the option to publish the peer review history of their article (what does this mean?). If published, this will include your full peer review and any attached files.

Reviewer #1: No

Reviewer #2: No

---

## [Author Response · Author response to Decision Letter 0]

12 Jan 2020

Additional Editor comments:

Major points:

-The intrinsic types of breast cancer are not defined according to prior studies (PMID: 29132568).

Thank you for this comment. As we mentioned in the section “Surrogate Definition of intrinsic subtypes” expression of hormone receptors, HER2 and analysis of Ki67 is generally used to define the intrinsic subtypes in breast cancer. Unfortunately, Ki67 was not available in our database. Therefore, the distinction between Luminal A and B was a challenge. Based on the publications of Minckwitz et al., Parise et al. and Lips et al. we used grading as a surrogate parameter for Ki67 as we did in all our previous BRENDA publications. 

Anyway, the analysis and interpretation of the marker Ki67 is complex as some studies have used 10%, 14% or 20% as cut-off value. This problem is also discussed in PMID: 29132568 you mentioned.

-Fisher exact test should be used in some subgroups' analyses, instead of Chi-square test.

As we all know, the usual rule of thumb for deciding whether the chi-squared approximation is good enough is that the chi-squared test is not suitable when the expected values in any of the cells of a contingency table are below 5, or below 10 when there is only one degree of freedom. In our case we have no cells with an expected frequency of less than 5. Therefore, the chi-square test is good enough.

Nevertheless, we additionally calculated Fisher’s exact test beside of two other tests (likelihood ratio and linear by linear association). 

The asymptotic significance of the chi-square test and the exact significance of Fisher’s exact test are in our case identical at least to 3 decimal places. Therefore, we renamed “significance” in table 1 “basic characteristics” to “asymptotic and exact significance”.

Here are the results

Age groups

T categories

Grading

Nodal status

Intrinsic subtypes

100% guideline adherence

Guideline adherent breast conserving therapy (BCT)

Guideline adherent radiotherapy

Adjuvant systemic therapy (AST)

-Please compare the characteristics of the subgroups that were generated by PSM-IPW grouping. I.e. to test whether PSM-IPW worked.

Thank you for this very interesting comment. We compared the characteristics of the subgroups that were generated by PSM-IPW grouping and indeed, in our case PSM-IPW increases the imbalance between the groups. This is just what Gary King and Richard Nielsen (2018) stated in their publication “Why Propensity Scores Should Not Be Used for Matching” (although propensity score matching is an enormously popular method of preprocessing data for causal inference) [27]. They showed that “PSM often accomplishes the opposite of its intended goal — thus increasing imbalance, inefficiency, model dependence, and bias. The weakness of PSM comes from its attempts to approximate a completely randomized experiment, rather than, as with other matching methods, a more efficient fully blocked randomized experiment. PSM is thus uniquely blind to the often large portion of imbalance that can be eliminated by approximating full blocking with other matching methods.”

Therefore, we no longer use PSM-IPW. We test now if there is a difference between two or more survival curves using the Gp family of tests of Harrington and Fleming (1982), with weights on each death of S(t)p, where S(t) is the Kaplan-Meier estimate of survival. With 

rho = 0 this is the log-rank or Mantel-Haenszel test, and with rho = 1 it is equivalent to the Peto & Peto modification of the Gehan-Wilcoxon test. As result, there is no significant difference (p=0.7) between the survival curves of low risk patients with radiation and those without radiation.

-How many of your patients probably have used gene based biomarkers (Oncotype or mammoprint)? As recent trials show, they may influenced the patient treatment and outcomes.

At the time of data collection between 2001 and 2009 multigene expression assays were not the standard of care. So, for this retrospective analysis we do not know if individual patients received biomarker testing. But we agree that for further studies there will be subgroups of patients in which the multi gene expression assays will provide further helpful information and will also influence the patient`s decision on treatment strategy. 

Minor points:

-Please cite related articles (PMID: 29132568, PMID: 28746732 and PMID: 31451977)

We cited the related articles

-Please indicate in the main text, not in the supplement, that the ethical approval has been reviewed and obtained.

We indicated that ethical approval has been obtained in the main text. 

Reviewer #1: You state” The treatment of elderly breast cancer patients differs from the therapeutic approach in younger ones, as elderly patients are prone to geriatric frailty and comorbidities, such as renal failure, 94 liver disease, and/or cerebrovascular disease (3-5).”

Please add that the treatment also changes because the cancers are less aggressive (Even within ER +, elderly seem more responsive to endocrine therapy as the percent ER positivity is higher.) 

Thank you for this important comment. We added in the discussion section that therapeutic strategies in elderly patients often differ from those of younger ones due to less aggressive tumor types. 

You over state the benefits of RT:

“After evaluation of different subtypes of breast cancer, it is obvious that even those with less aggressive molecular types profit from radiotherapy. In different meta analyses, hazard ratios of 0.2-0.35 have been described meaning that up to eight of ten relapses can be avoided by RT (6-8). As a result of the improved local tumor control, it has been previously stated that breast cancer related mortality has been significantly decreased during the last decades (6). Therefore, it has been a general recommendation to perform BCT only in combination with postoperative radiotherapy (9). If patients cannot or will not accept postoperative RT, mastectomy is the alternative surgical treatment (10). However, subgroups have been defined that are associated with a low risk for recurrence (pT1, pN0, R0, HER2-) (11, 12). In those cases, radiotherapy still adds benefits but can most likely be dispensed even after BCT.”

No randomized trial has shown a survival benefit to RT in early breast cancer. 

In the cited meta-analysis by the EBCTCG 17 randomized trials were analyzed and showed that RT to the breast reduced the breast cancer death rate by about a sixth. 

Always state “locoregional” recurrence, as RT has no impact on distant recurrence in randomized trials. 

Thank you for this comment. We have changed the section to make clear that we talk about locoregional recurrence. 

You never defined intermediate vs high risk or separated out those groups. These need to be defined.

In fact, we revised the manuscript and only have 2 groups now: low-risk patients who do not need additional radiation therapy and patients at higher-risk who seem to need radiation therapy. 

You combine intermediate and high risk into a single group without showing that they act similarly. You need to describe why you made this combination. When we analyzed our data we found that the major differences occurred between low-risk patients and others. As mentioned above we no longer differentiate between intermediate and high risk patients.

In some places you state high risk benefits and in others that intermediate or high risk benefits, but I do not see these differentiated in your data. 

Thank you for this comment. We adjusted the article accordingly. We no longer differentiate between intermediate and high risk patients, instead we use the term higher risk. 

Your intermediate group likely contains patients that fit the randomized trials that showed RT has not beneficial. (Such as a 1 cm Grade 3 cancer.). If you are using retrospective data to invalidate the conclusions of a randomized trial, please state why you think this is defensible. 

We would really like to correct this issue since we don`t try to use retrospective data to invalidate the conclusions of the randomized trial CALGB 9343. Hughes et al. included patients >69 years with T1N0 ER+ breast cancer only. In this trial they did not differentiate between intrinsic subtypes. They found a small decrease in IBTR but no significant benefit in survival. 

When you state that limiting your analysis to 9343 criteria, are you including intermediate and high risk patients who meet that criteria? If so, does that negate your statement that intermediate and high risk need RT? 

Thank you for this interesting comment. A similar question can be asked for CALGB 9343. A better way to think of risk is as the possibility or probability of recurrence or death. Therefore, patients at higher risk without RT have a higher probability of recurrence or death compared to low risk patients. Nevertheless, there are patients at higher risk without RT, who have no recurrent disease. In the CALGB 9343 trial, as well as this study two types of low risk patients are defined, which both don´t need RT. Therefore, we can combine these two subsets to a new subset of low risk patients. For this combined subgroup we have no significant difference in recurrence free survival (log rank (Mantel-Cox) p=0.958) and tumor associated survival (log rank p= 0.506)

Reviewer #2: The manuscript here is technically sound and adds additional important information to the literature using a relatively large database of elderly women with breast cancer. The manuscript could be simpler and clearer and my suggestions are below:

1. The use of the terms BCT with and without RT is confusing. If I’m not mistaken breast conserving therapy without radiotherapy should be referred to as BCS (breast conserving surgery). BCT implies the use of radiotherapy. 

Thank you very much for this important issue. We have revised the terms. 

2. I generally disagree with intro lines 102-104. There has been enough data on low risk patients with small tumors having low recurrence rates without radiotherapy that these statements do not apply in 2019. 

Thank you very much for this comment. We have clarified the introduction. 

3. I’m not sure why the authors separated the groups out by low and intermediate/high. They define the patients that they place in this intermediate/high group but never define the difference between intermediate and high. This should be defined at some point. 

Since reviewer 1 discussed the same issue we would kindly refer to the statement further above.

4. There are a great deal of words in the paragraph called “basic characteristics” and it is a difficult read. It is essentially explaining Table 1. The paragraph could be started by simply stating that “Basic patient characteristics are listed in Table 1” and then just point out a couple of important distinctions. I do not feel this large paragraph is necessary. 

We have revised the section in order to shorten the paragraph. 

5. I find the terms throughout the manuscript of guideline adherent and guideline non-adherent confusing. One could simply state that guidelines state that radiotherapy is indicated and compare the difference between those having it and those not having it. It also makes the Figures showing relapse free and overall survival confusing. 

Thank you for this comment. We think that it is quite interesting to see in how many cases RT was not carried out even though the national guidelines would have recommended to do (at least 27,3 %!). Around 30% of those were of intermediate- and high-risk tumors and we showed that those patients profited from RT. 

6. Discussion could be much simpler. Prime 2 and CALGB trials are referenced early on and then a paragraph is allotted to each one later in the discussion. I would just acknowledge each randomized control trial once and shorten discussion. 

We adjusted the discussion according your recommendation. 

7. Overall useful information particularly regarding the higher risk group but could be more simply and clearly presented. 

We tried to reword the manuscript in order to achieve a clearer presentation.

---

## [Decision Letter · Decision Letter 1]

10 Feb 2020

Effect of adjuvant radiotherapy in elderly patients with breast cancer

PONE-D-19-28656R1

Dear Dr. Stueber,

We are pleased to inform you that your manuscript has been judged scientifically suitable for publication and will be formally accepted for publication once it complies with all outstanding technical requirements.

With kind regards,

Lanjing Zhang, MD, MS

Academic Editor

PLOS ONE

Additional Editor Comments (optional):

Reviewers' comments:

Reviewer's Responses to Questions

**Comments to the Author**

1. If the authors have adequately addressed your comments raised in a previous round of review and you feel that this manuscript is now acceptable for publication, you may indicate that here to bypass the “Comments to the Author” section, enter your conflict of interest statement in the “Confidential to Editor” section, and submit your "Accept" recommendation.

Reviewer #1: All comments have been addressed

2. Is the manuscript technically sound, and do the data support the conclusions?

Reviewer #1: Yes

3. Has the statistical analysis been performed appropriately and rigorously? 

Reviewer #1: Yes

4. Have the authors made all data underlying the findings in their manuscript fully available?

Reviewer #1: Yes

5. Is the manuscript presented in an intelligible fashion and written in standard English?

Reviewer #1: Yes

6. Review Comments to the Author

Reviewer #1: Revisions seem correct to me at this time in reading these comments. Thank you for your revisions This seems fine

7. PLOS authors have the option to publish the peer review history of their article (what does this mean?). If published, this will include your full peer review and any attached files.

Reviewer #1: No

---

## [Editor Report · Acceptance letter]

9 Apr 2020

PONE-D-19-28656R1 

Effect of adjuvant radiotherapy in elderly patients with breast cancer 

Dear Dr. Stueber:

I am pleased to inform you that your manuscript has been deemed suitable for publication in PLOS ONE. Congratulations! Your manuscript is now with our production department. 

With kind regards,

on behalf of

Dr Lanjing Zhang 

Academic Editor

PLOS ONE